# Detection of Microplastics in Water and Ice

**Seohyun Jang** [1], **Joo-Hyung Kim** [1,2] **and Jihyun Kim** [2,*]

1.  Laboratory of Intelligent Device and Thermal Control, Department of Mechanical Engineering, Inha University, Incheon 22212, Korea; 22201200@inha.edu (S.J.); joohyung.kim@inha.ac.kr (J.-H.K.)
2.  INHA Institute of Space Science and Technology, Inha University, Incheon 22212, Korea
*   Correspondence: jihyunn.kim@inha.ac.kr

**Abstract:** It is possible to detect various microplastics (MPs) floating on water or contained in ice due to the unique optical characteristics of plastics of various chemical compositions and structures. When the MPs are measured in the spectral region between 800 and 1000 nm, which has relatively little influence on the temperature change in water, they are frequently perceived as noise or obscured by the surrounding reflection spectra because of the small number and low intensity of the representative peak wavelengths. In this study, we have applied several mathematical methods, including the convex hull, Gaussian deconvolution, and curve fitting to amplify and normalize the reflectance and thereby find the spectral properties of each polymer, namely polypropylene (PP), polyethylene terephthalate (PET), methyl methacrylate (PMMA), and polyethylene (PE). Blunt-shaped spectra with a relatively large maximum of normalized reflectance ($NR_{max}$) can be decomposed into several Gaussian peak wavelengths: 889, 910, and 932 nm for the PP and 898 and 931 nm for the PE. Moreover, unique peak wavelengths with the meaningful measure at 868 and 907 nm for the PET and 887 nm for the PMMA were also obtained. Based on the results of the study, one can say that each plastic can be identified with up to 81% precision by compensating based on the spectral properties even when they are hidden in water or ice.

**Keywords:** microplastics; near-infrared; convex hull; Levenberg–Marquardt curve fitting; identification

## 1. Introduction

Global plastic production has increased every year since 1950, reaching 368 million tons in 2019 [1–4]. This is because of the attractive properties of plastics, such as lightweight, durability, versatility, and low cost of production that led to their large-scale use in both households and industries [5–7]. Uncontrolled large and small plastic waste released into the environment is about 12 million tons per year and has emerged as one of the environmental pollution issues due to the plastic's low degradability and recycling rate coupled with its high consumption [2,4,8]. The commonly found plastics include polyethylene (PE), polypropylene (PP), polyethylene terephthalate (PET), methyl methacrylate (PMMA), ethylene-vinyl acetate (EVA), polystyrene (PS), and polyamide (PA), which, when disposed of, become smaller and smaller pieces over time by processes, such as the continuous motion of the debris and ultraviolet radiation [9,10]. Then, the plastic particles or synthetic organic polymers having a size between 100 nm and 5 mm are called microplastics (MP) [9–12]. They are rather difficult to detect because of their small size and low concentration in various environments, including sea [11,13–17], lakes [18], rivers [19], soil [20,21], and so on [2,7]. Additionally, plastics such as PE and PP mostly float on the water, since they have low densities of 0.855 to 0.946 g/cm$^3$ and 0.88 to 0.96 g/cm$^3$, respectively [22–24], while at the same time, sediments or ices also act as sinks for MPs [6,7,9,10,25]. Although large amounts of floating or suspended plastic debris are observed in water or ice, a sufficient number of comprehensive analyses and monitoring tools for the sources, sinks, pathways, and type or abundance of debris are not available [24,26]. Particularly, there exists insufficient monitoring of the sources and routes of the MPs, such as the PP, PE,

PMMA, and PET, which are mainly detected in the water of semi-enclosed bays and coastal zones of urban and rural areas in South Korea [23].

The analytical methods using the near-infrared (NIR, 750 to 2500 nm) spectroscopic region are fast, non-destructive, non-invasive, and nearly universally applicable and require only minimal sample preparation [27,28]. The NIR spectra are chosen as the predominant wavelengths in remote sensing, representing these radiances as a ratio that minimizes atmospheric influence and differences in sensor geometry without greatly complicating the data processing [29,30]. The spectral properties of the polymers in the NIR have been conventionally used in the automatic optical sorting of wastes in the recycling industry and are used to identify environmental MPs with direct and indirect monitoring approaches such as remote sensing [24,27,31–34]. This is because of the fingerprint regions of the polymers, such as C-H, O-H, N-H, C-C, and C-O absorption bands that are easily observable [24,26,27,31–34]. To interpret the remote sensing measurements of the MPs floated on water or contained in ice, the spectra of the polymers are distinguished from those of the water. The absorption bands of water and ice in the NIR had been reported as 760, 970 (or 980), 1190, 1450, and 1940 nm for liquid water and 1030 nm for water ice [35,36]. The water and ice absorbance in the NIR range increases or the peak wavelengths shift with a temperature increase, but the change is relatively small in the wavelength region between 750 and 1000 nm [37–39]. The absorption properties of water result from the vibration transition in $H_2O$, involving various harmonics and combinations-induced asymmetric and symmetric stretch and bending mode transition by H-O-H [40]. The dominant absorption mechanisms for ice are the molecular vibrations in the NIR region, and the corresponding absorption is stronger than the visible region [41–43].

Based on the characteristic spectral properties of plastics, water, and ice, this study focuses on detecting the type of MPs floated on water and placed inside and outside ice with a range of 800~1000 nm, by coding that followed several mathematical techniques, such as the continuum-removal process and Levenberg–Marquardt (LM) algorithms [29,30]. The present detecting method will be applied to trace the sources and pathways of plastics in the waters of urban and rural areas in South Korea.

## 2. Experimental Materials and Methods

### 2.1. Preparation of Samples

The four different types of plastics that are not colored but are of various opacities are divided into four categories as shown in Figure 1. The PP, PET, and PMMA are manufactured by Ilwoong Platech, and the PE is produced from QTech of South Korea. The sizes of the plate-shaped samples are $50 \times 50$ mm$^2$ and $5 \times 5$ mm$^2$, and they have a rectangular cross-section with 5 mm thickness that excludes the PET (which is 3 mm in thickness). Three pieces of each smaller plastic are used for single-species detection, and one piece each is used to sort each plastic in the plastic mixture. All of these are put in vials of diameter 40 mm and height 3 mm. The vials are filled with 3.8 mL distilled (DI) water. The amount of DI water and the volume of the vial are chosen to minimize the movement of the floating plastics. The DI water is used to reduce the measurement error coming from impurities. The spectra measured from the plastic polymers floated on the DI water and shown in Figure 1d are considered as the reference to figure out the variation of spectra intensities and the main wavelengths of plastics embedded into ice as shown in Figure 1b,c. Then, the bottled DI water and plastics are maintained in a cooler at 20 °C temperature for 24 h, and the ice covering the plastics is thick enough to be observed with naked eyes easily. The temperature of water and ice are kept at 35 and 0 °C, respectively, to minimize the change in absorbance due to the temperature change.

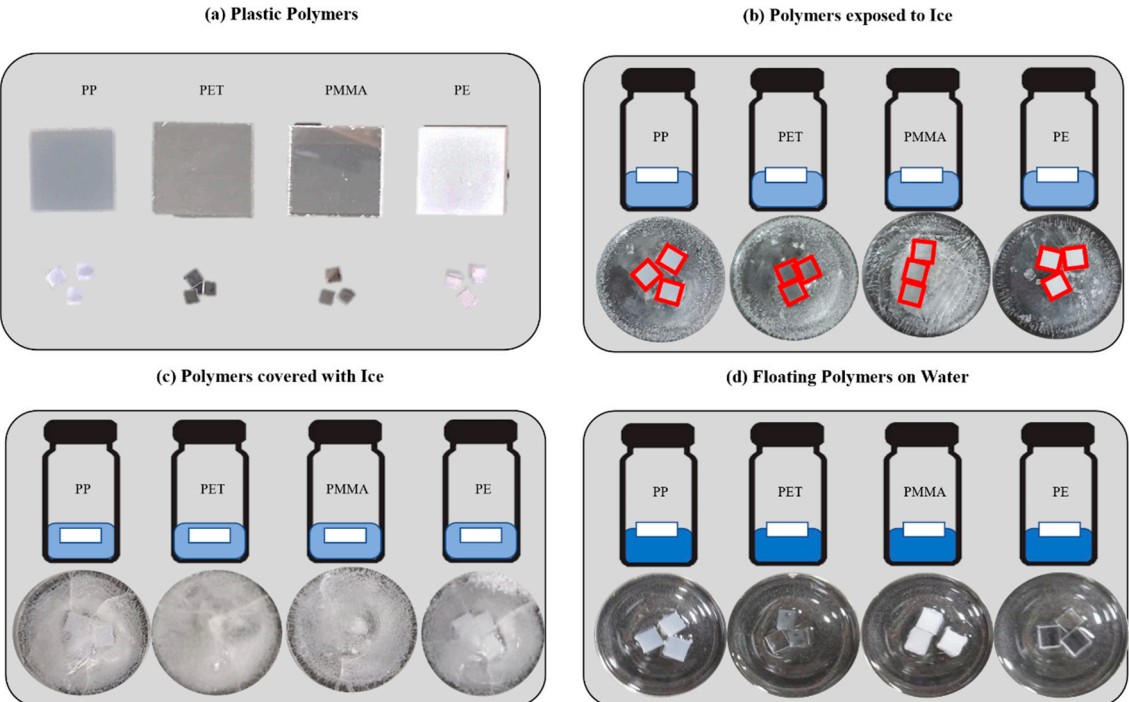

**Figure 1.** Schematic and pictures of sample materials. (**a**) Plastic polymers, (**b**) plastic polymers exposed to ice, (**c**) plastic polymers covered with ice, (**d**) floating plastic polymers in water.

### 2.2. Experimental Setup

The spectral reflectance of the plastic polymers under laboratory conditions was measured in a black room with an ASD FieldSpec 3 spectrometer (Analytical Spectral Devices Inc., Boulder, CO, USA) in a wavelength range of 350 to 2500 nm, as shown in Figure 2. This equipment combines three spectrometers to convert the solar reflected portion of the spectral region, with a sampling interval of 1.4 nm for the $350 \sim 1000$ nm region and 2 nm for the $1000 \sim 1800$ nm and $1800 \sim 2500$ nm regions [44]. In the considered NIR range of $800 \sim 1000$ nm, the spectrometer can measure all measurable wavelengths with a spectral resolution of 3 nm. A 1250W halogen lamp, H25s (HEDLER Systemlicht, Runkel, Germany), was used as the illumination source with a degraded output of almost 60% of the original power. It was positioned at a distance of 57 cm from the sample at an incident angle of 35° to reduce the noise from the light source. The signal conditioning components (SCC) and data acquisition (DAQ, SC-2345) are connected simultaneously to control and maintain the output power of H25s. A 99% Spectralon panel ($25.4 \times 25.4$ cm², SRT-99-100, Labsphere Inc., North Sutton, NH, USA) was used as a white reference to compute the reflectance values. The measured radiance values were transformed into spectral reflectance, which is equal to the ratio between the radiance reflected by the plastic polymers and the one from the standard white reference plate [45]. A reference spectrum under the same measurement conditions was recorded as the average of 10 scans carried out continuously before the first measurement, to minimize the noise level in the spectral signal. Then, the samples are placed on the 99% white Spectralon panel vertically apart from an 8° fiber optic of the ASD spectrometer. The distance between the reflectance target and the fiber optic for plastic polymers is kept at 10 cm to get clear reference data at a close range. However, the MPs were placed inside vials (2 cm base diameter) detected at a distance of 28 cm, and the setup is capable of measuring a range of 14.43 cm² at a time. The distance between the samples and the detector is determined by the largest plated plastic polymers. The spectral reflectance measured by the ASD optics is transmitted using wireless fidelity (Wi-Fi) communication to the RS3 software for real-time spectral analysis. A laser that has a typical wavelength of 650 nm is kept above

the ASD fore optic and was used to calibrate the shifted signals in each detection trial and to determine the best position to place the sample on the white reference plate. The entire experimental setup, except the front side, is covered with the blackout curtain to eliminate external influences. To achieve necessary analytical precision, the spectrometer measures the samples 10 times continuously and uses the average values in the wavelength of 800 to 1000 nm.

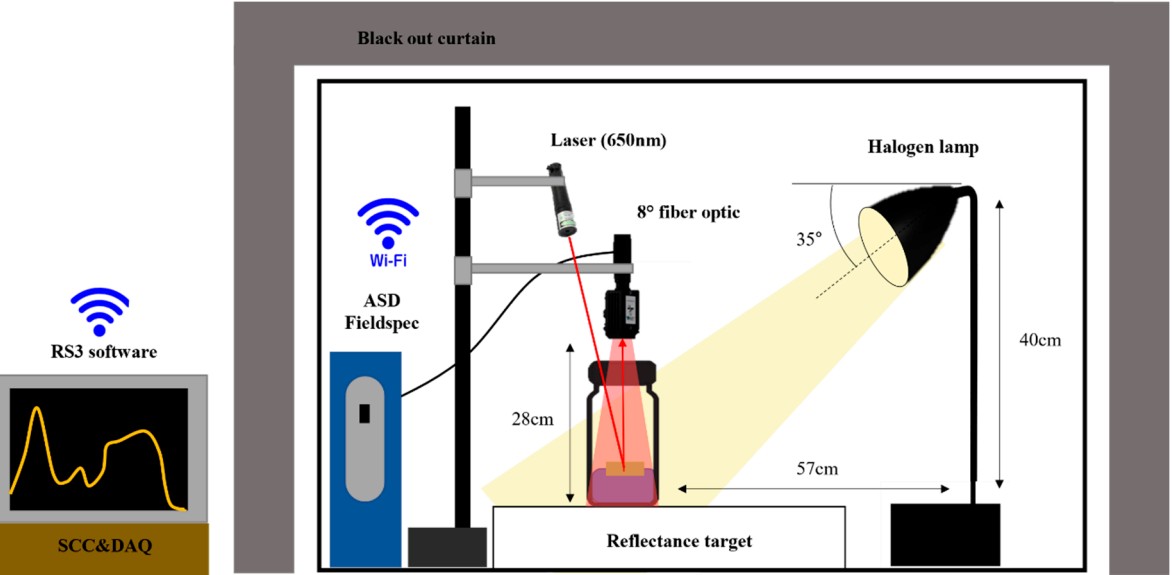

**Figure 2.** Schematics of the experimental setup (SCC: signal conditioning components, DAQ: data acquisition).

### 2.3. Analysis of Spectra

A spectrum derivate analysis is a typical technique that has been used to identify absorption features in floating materials and other optically active constituents of natural waters and ices [24,46–48]. The continuum-removal process isolates the spectral features from the combined spectral signals and positions them at the same level, allowing the mixed spectra to be compared with each other. The continuum removal and the feature comparison processes are successful spectral identification methods applied with weak or indistinguishable spectra [49]. Additionally, a relative band depth index algorithm derived from a continuum line quantified major absorption feature [24,49] is also used in the analysis. It allows for spectral deconvolution by converging the spectrum to zero. End and start points of the continuum line are formed by an app, OriginPro 2020 (Academic) 2D Convex hull (v.1.0), to systematically select a convex hull [24]. Figure 3a,b show how to perform the convex hull based on the raw data and rearrange the spectrum for further analysis, respectively. The LM algorithm is the most widely used method to solve nonlinear curve fitting problems in spectral mixed models. The signals of the spectrometer, mixed with several peaks and expressed in a blunt single curve, can be separated into several Gaussian functions with a single peak by the LM algorithm. It is based on a nonlinear least square method of Equations (1) and (2) and changes a variable matrix of $p$ in a direction that minimizes errors $E(p)$ between the real data and curves fitted data using a rotating iteration [50].

$$E(\boldsymbol{p}) = \sum_{i=1}^{n} \left[\boldsymbol{r}_i(\boldsymbol{p})\right]^2 = [\boldsymbol{r}_1(\boldsymbol{p}) \cdots \boldsymbol{r}_n(\boldsymbol{p})] \begin{bmatrix} \boldsymbol{r}_1(\boldsymbol{p}) \\ \vdots \\ \boldsymbol{r}_n(\boldsymbol{p}) \end{bmatrix} = \boldsymbol{r}^T \boldsymbol{r} \tag{1}$$

$$\boldsymbol{r}_i(\boldsymbol{p}) = \sum_{i=1}^{n} [y(x_i) - \hat{y}(x_i; \boldsymbol{p})], \ (1 \leq i \leq n) \tag{2}$$

where $y(x_i)$ is the measured data, and $\hat{y}(x_i; p)$ is the curve fitted $\hat{y}$ calculated using $p$ for the same $x_i$, $1 \leq i \leq n$. $r$ is the column matrix sets of $r_i(p)$, which is the residual between $y(x_i)$ and $\hat{y}(x_i; p)$.

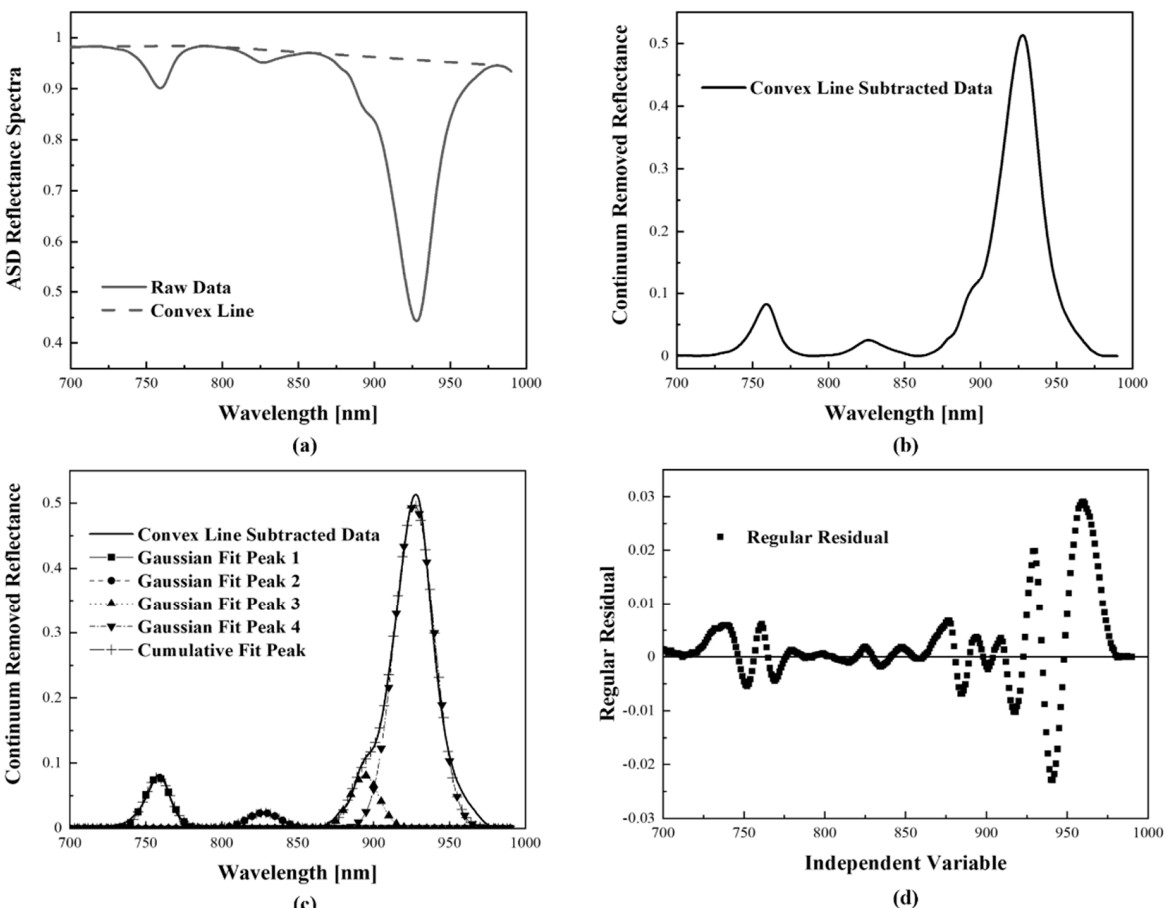

**Figure 3.** The spectral analysis of PE in $50 \times 50$ mm$^2$ (OriginPro 2020 (Academic)). (**a**,**b**) Continuum removed with the 2D convex hull; (**c**,**d**) LM algorithm with peak deconvolution.

Gaussian–Newton

$$p_{k+1} = p_k - \left(J_r^T J_r\right)^{-1} J_r^T r(p_k), \ k \geq 0 \tag{3}$$

Gradient Descent

$$p_{k+1} = p_k - 2\lambda_k J_r^T r(p_k), \ k \geq 0 \tag{4}$$

Levenberg–Marquardt

$$p_{k+1} = p_k - \left(J_r^T J_r + \mu \, diag\left(J_r^T J_r\right)\right)^{-1} J_r^T r(p_k), \ k \geq 0 \tag{5}$$

Various methods have been employed to reduce the residual and consequently minimize the $E(p)$, among which the LM method (Equation (5)) is a combination of the Gaussian–Newton and the gradient descent methods (Equations (3) and (4)). This method, hence, enables a rapid and stable convergence even if the initial value is far from the solution [51–53]. The Jacobian matrix $J$ of $r$ is defined as $J_r = \frac{\partial r_k}{\partial p_k}$, $1 \leq k \leq n$. Additionally, the term subtracted from $p_k$ in the expression of each method is $\delta p$. This value is adjusted from the previous value of $p_k$ so that $p_{k+1}$ can reduce errors with the real data [54]. The value corresponding to $\delta p$ in the coding is mainly calculated for the changing $p$, and the $E(p)$ at that time is measured to adjust $\mu$, the damping parameter. The Gaussian–Newton

method is used when it is decided to converge steadily towards the solution with a $\mu$ that is smaller. However, if the current iteration is far away from the solution with a larger $\mu$, the system can be quickly brought closer to the solution using the gradient descent method [55]. Thus, $\mu$ is adjusted as the procedure is repeated so that it accurately converges even on complex graphs and can respond quickly to highly variable data, such as the one from real environments. The $p$ converging to the solution by the LM method is, therefore, replaced by the main constants of the Gaussian function and disassembles the first graph into several Gaussian functions. Figure 3c describes an example of a convex line subtracted data (black line) that is disassembled into four different Gaussian functions. Each Gaussian function has one single peak, and the sum of them has resulted in the cumulative fit peak (cross-shaped line). The difference between the initial signal and the cumulative fit peak is the residual $r_i$ of Equation (1) and is shown in Figure 3d.

## 3. Result and Discussion

### 3.1. Peak Wavelengths of the Plastic Polymers

The peak wavelengths of the PP, PET, PMMA, and PE, of size $50 \times 50$ mm$^2$ with a rectangular cross-section and 5 mm thickness, were measured using the Lambda 750 UV/Vis/NIR spectrometer (PerkinElmer, USA). These measurements are carried out in the range of 800 to 1000 nm with the following wavelengths for the plastics: (1) 920 and 927 nm for the PP, (2) 873 and 914 nm for the PET, (3) 903 nm for the PMMA, (4) 933 nm for the PE. The distinguishable absorption bands are compared with the results of Garaba et al. [24] as described in Table 1. Garaba et al. [24] experimented on marine-harvested MPs, which showed that each plastic has unique absorption peak wavelengths because of the different molecular structures of plastics. The absorption peak wavelengths of the PP and PE seem to be almost alike, and those of the PET and PMMA have alike waveforms but are slightly shifted or not measured. Green et al. [35] examined the absorption spectra of water and ice to reveal interesting facts, which include different spectral signals for different phases of the material, even though the molecular structure of the material is the same. Since these factors fulfill the primary role in interpreting the conditions mixed with numerous signals, the NIR spectra region, 800 to 1000 nm, is selected to identify the plastic polymers.

**Table 1.** Representative peak wavelengths for polymers used in this study.

| Material | Wavelength (nm) | Normalized Absorption | Material | Wavelength (nm) | Normalized Absorption |
|---|---|---|---|---|---|
| PP | 911 * <br> 935 * | 0.414 <br> 0.437 | PMMA | 897 * | 0.779 |
| PET | 863 * <br> 899 * | 0.280 <br> 0.284 | PE | 929 * | 0.337 |

*: similar/same peak wavelengths of Garaba et al. [24] and this work.

### 3.2. Coding for Plastic Polymer Classification

The coding (Supplementary Materials) to analyze the observed spectra gained from the various conditions is developed with Microsoft Visual Studio Community 2019 (Ver. 16.4.5) and the convex hull, continuum-removal process, and LM algorithms were used to identify the plastic polymers. This development of software consists of three parts as shown in Figure 4: input (Part 1), analysis of spectrum (Part 2), and the result (Part 3). All initial settings such as the reference peak, file name, spectral range, input raw data files, and environmental conditions are performed by the users in the Part 1 stage itself. Based on the reference value, for instance, 654 nm, the input raw data are consecutive rearranged, averaged, and plotted in Part 2. A set of raw data can be dragged and dropped into the "Data" section, and the average in the user-defined range is used for the analysis. The "Environment" includes several conditions such as default, water, and ice under which measurements were performed. In the ice or water mode, the compensation process is carried out to remove the specific Gaussian function due to the environmental factors, and

a re-curve fit is performed to modify the spectrum. The spectrum of "PP with water" is analyzed through the compensation process that excluded distinct waterborne areas and predicted the polymers. The convex hull, continuum line subtraction, decomposition with multiple Gaussian functions, and LM curve-fitting in Part 2 are processed step-by-step for the users to monitor the comprehensive analysis. Using the first derivative method, the visible peak wavelengths are automatically found or added on the chart by the users and are converted into multiple Gaussian functions giving the initial value for the LM method. After LM fitting, the compensation can only be performed for the ice and water, which removes the signals of water and ice and can identify the plastic polymer easily by their spectra. In the process of using the identification algorithm, the peak wavelength of each substance can be classified as an accordant polymer if the difference of the wavelength is less than ±3.5 nm based on the optical characteristics of the pure polymer. The peak wavelengths of pure polymers are given in Table 1. The discrepancy is due to environmental factors with higher reflectance that cause the wavelength to shift. The classified peak wavelengths represent the appropriate polymer, and the non-classified peaks are included in the category, "None" of the graph and "Detected peak", in Part 3.

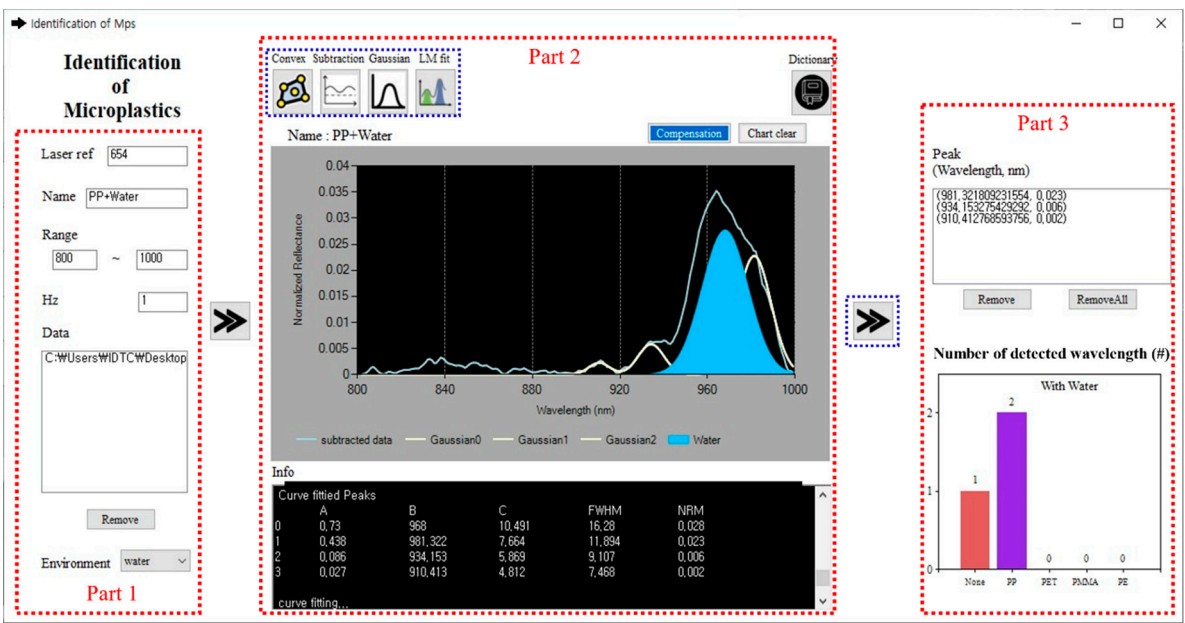

**Figure 4.** Image of the in-house program for microplastic classification. For instance, the water is selected in the 'Environment' in Part 1, and the classified peak wavelengths in Part 2 represent one non-identified peak (None) and two identified peaks (PP) in Part 3.

### 3.3. Optical Characterization of Plastic Polymers

The normalized reflectance spectra of the PP, PET, PMMA, and PE were measured with their scales by the ASD spectrometer (black lines). The spectral lines were nonlinearly fitted using the LM method with a tolerance of $10^{-6}$ through the continuum-removed process (red line) in the NIR spectral range of 800 ~ 1000 nm, as shown in Figure 5. At that time, the peak wavelength, $\lambda_p$, maximum of normalized reflectance ($NR_{max}$), and full width half maximum (FWHM) were measured to be used as optical characterization indices for the Gaussian function. The $NR_{max}$ is the intensity of the corresponding peak wavelength in the reflection, and the higher this value, the clearer the detection of the peak wavelength in the mixed spectra. The peak wavelength of 897 nm of the PMMA was not measured in the previous research because it is difficult to distinguish it in its overlapping with the peak wavelengths of other polymers in the mixed plastics even though the $NR_{max}$ is large. The wavelength that assumes a value of around 903 nm is detected clearly from

the UV/VIS/NIR spectrometer. This trend is similar to the one for other materials. Some of the peaks slightly shift from the previously measured values.

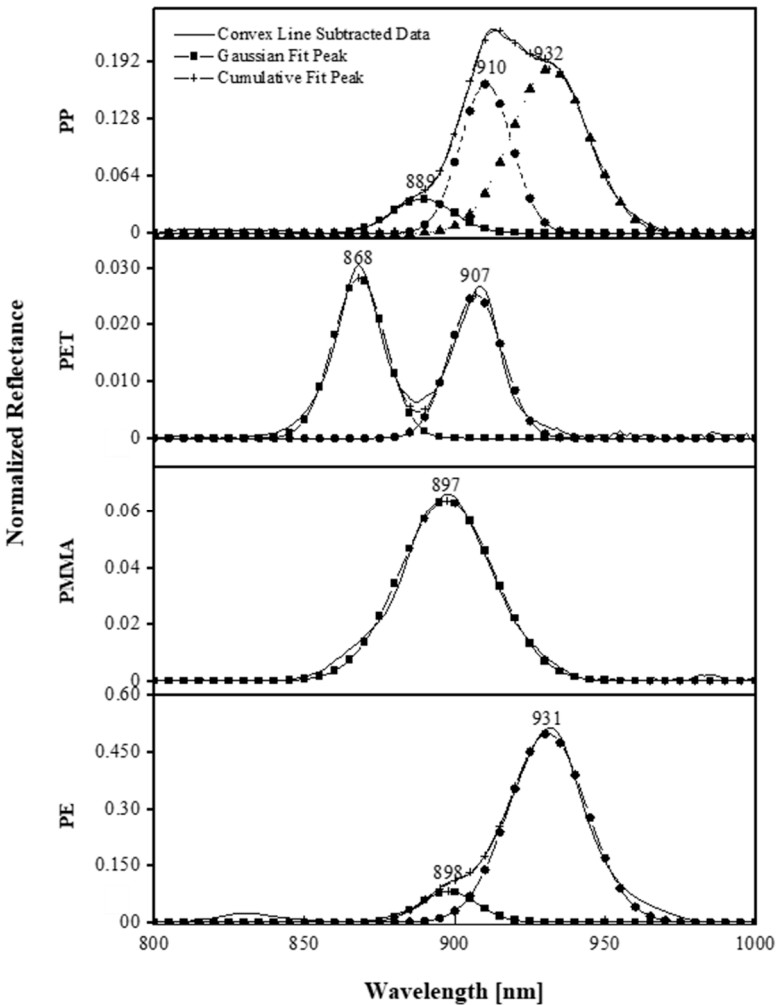

**Figure 5.** The normalized reflectance of the PP, PET, PMMA, and PE, which shows the characteristic peaks of the material and the core components of the Gaussian functions of plastic polymers ($\lambda_p$: Gaussian peak wavelength, $NR_{max}$: maximum of normalized reflectance, FWHM: full-width half maximum with a unit of nm ).

The PP and PE show the optical property of a blunt single signal combined with two or three different Gaussian peak wavelengths, respectively, and have larger $NR_{max}$ values. Being able to minimize noise, which is an advantage of remote sensing in the NIR region, allows the mixed peaks to be decomposed in high resolution even though they are in a narrow range. The wavelength band of PP, $\sim$ 932 nm, is not visible in the mixture of the PP and PE because its $NR_{max}$ is about one-third smaller than that of the peak wavelength of PE, which is $\sim$ 931 nm. Thus, the respective peak wavelength, 910 nm and 898 nm, of PP and PE can play a possible role in the classification of plastics. The characteristic peak wavelengths of PET can be located at 868 and 907 nm. The peak wavelength of 907 nm when it is overlapped with other plastic polymers such as the PE and PP makes it difficult to distinguish the PET polymer because of the smaller $NR_{max}$ value. Nevertheless, it can be identified with two wavelengths considered as a criterion. Therefore, the materials are distinguished with the peak wavelengths: 889, 910, and 932 nm for PP, 868 and 907 nm for PET, 897 nm for PMMA, and 931 nm for PE.

### 3.4. Optical Properties of Plastic Polymers with Ice and Water

It is sometimes challenging to distinguish the transparent or milky-white colored plastic polymers when they are exposed to ice or while floating on water. The peak wavelengths of ice and water in this experiment are located at 894 and 963 nm, respectively, after extracting the spectral lines fitted using the LM method as shown in Figure 6. In the case of water, the peak wavelength was shown at 970 nm, similar to the previous study, but it somewhat shifts due to the process of breaking the spectral data around 1000 nm [35,36].

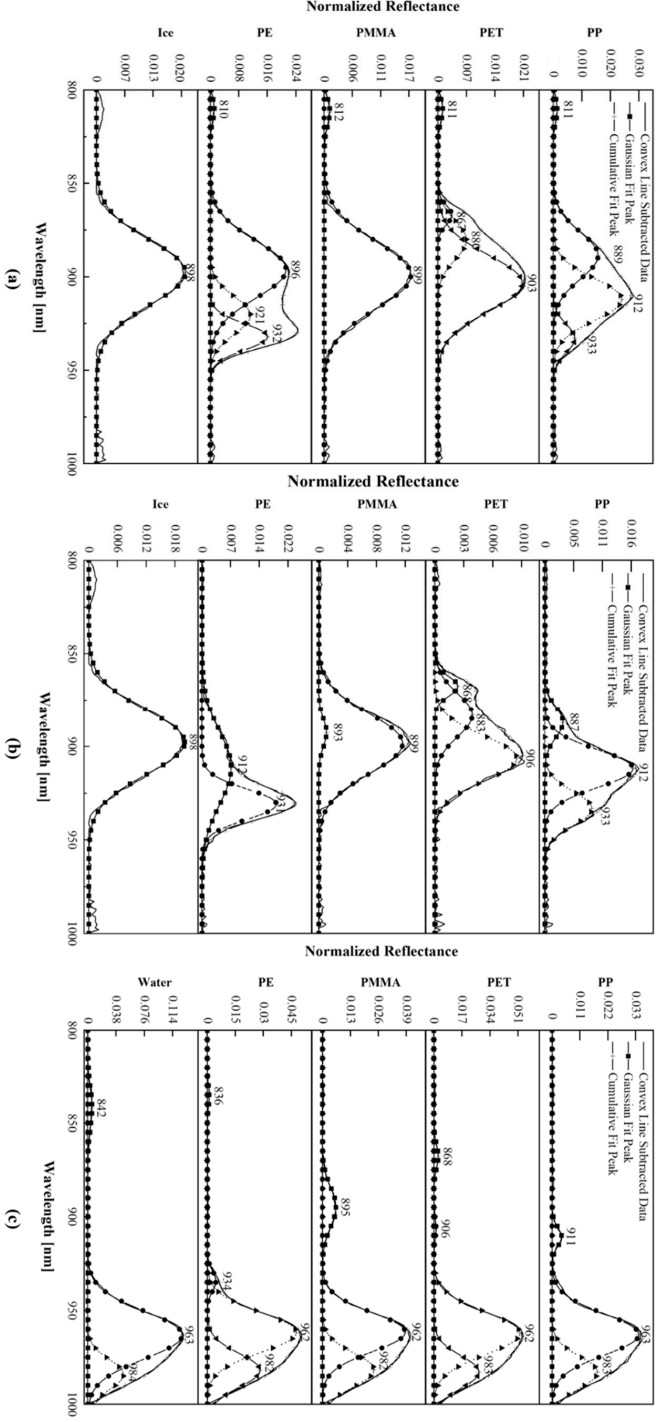

**Figure 6.** The spectral of the characteristic peak wavelength of plastic polymers (**a**) exposed to ice, (**b**) covered with ice, and (**c**) floating on water.

Figure 6a shows that each material exposed to ice, except the PE, cannot be differentiated from ice in terms of the optical characteristics due to similar peak wavelengths and reflectance values measured. The noise signals measured by the scattered reflections from the water or ice are prominently observed in the PET with a small normalized reflectance value. However, it can extract meaningful spectral lines by using the LM method with Gaussian functions and distinguish the optical properties with the errors of $1 \sim 3$ nm. The PP and PE can be identified from the peak wavelength of the ice due to their larger $NR_{max}$. The peak wavelengths and normalized reflectance of the PET are very close to those of the ice but can be distinguished at 868 nm with an error of 3 nm by the LM method, whereas the PMMA with similar features is difficult to identity.

The majority of the peak wavelengths of the shallowly ice-covered plastic polymers are blended with that of ice and represented by a smooth and simplified curve so that it is hard to break them down into multiple Gaussian functions as shown in Figure 6b. The ice that overlaid them causes severe scattered reflections, resulting in some variations in the observed spectra. The spectral lines rendered by the LM method shift and the normalized reflectance also increases compared to the case of exposure to ice (Figure 6a). The materials can also be identified with peak wavelengths such as 912 nm for the PP, 868 nm for the PET, and 931 nm for the PE.

The spectra observed from plastic polymers floating on the DI water show a single peak wavelength with larger normalized reflectance at 963 nm and multi-peak wavelengths with the small one in other regions as shown in Figure 6c. The spectra of plastic polymers with that of water decomposed in them represent similar optical characteristics as displayed in Figure 5. The Gaussian peak wavelength of the PET and PMMA seem to be distinguished because they are far from the spectrum of the water, which is 963 nm, and are relatively less affected. The representative peak wavelengths with large $NR_{max}$ similar to the one for the PE can facilitate the detection of the plastic polymers under various environmental conditions. Even the PET with the smallest $NR_{max}$ can be distinguished as shown in Figure 6c when it does not overlap with the wavelengths of the water or ice. However, the decomposition of the Gaussian function can still result in peak shifts. The ice peak wavelength that is measured in the spectral range similar to that of the polymers and does not have an appreciably large $NR_{max}$ can be shifted in the presence of the polymers as described in Table 2. Then, it can measure at 912, 903, 899, and 896 nm the PP, PET, PMMA, and PE, respectively.

**Table 2.** The maximum of normalized reflectance $NR_{max}$ for representative peak wavelength of the plastic polymer in various environments.

| | Material Only | Exposed to Ice | Ice-Covered | Ice ($\lambda_{ice}$ = 898 nm) | Floating on Water | Water ($\lambda_w$ = 963 nm) |
|---|---|---|---|---|---|---|
| PP ($\lambda_{PP}$ = 932 nm) | 0.183 | 0.009 | 0.008 | 0.024 (912 nm) | 0.004 ($\lambda_{PP}$ = 910 nm) | 0.034 |
| PET ($\lambda_{PET}$ = 868 nm) | 0.028 | 0.002 | 0.003 | 0.021 (913 nm) | 0.003 | 0.052 |
| PMMA ($\lambda_{PMMA}$ = 897 nm) | 0.034 | 0.012 | 0.017 | 0.017 (899 nm) | 0.006 | 0.039 |
| PE ($\lambda_{PE}$ = 931 nm) | 0.498 | 0.019 | 0.016 | 0.021 (896 nm) | 0.005 | 0.048 |

### 3.5. Microplastics Classification in the Mixture

It can be aimed further to detect and classify each substance in the mixture to trace their sources and routes while measuring MPs remotely under various conditions. Figure 7 shows the degree of identification and classification of each plastic under various conditions with the use of the in-house program described in Figure 4. The x-axis represents the types of material and the y-axis indicates the number of detected wavelengths, between 800 and 1000 nm, of each plastic. Each plastic has been identified using its measured wavelengths through the compensation process. The numbers of peak wavelength ($N_{pw}$) identified by observing pure plastics are three for the PP, two for PET and PE, and one for PMMA (Figure 7a, black bar). As shown in Figure 7a, the plastics have been readily identified even

though $N_{pw}$ has decreased except for the PET when the single species has been contained in the water or ice. The PMMA, which has only one wavelength in the given region, cannot be identified even with increased precision through the compensation process because of their similar peak wavelengths when they are in the ice. Meanwhile, the process can identify the PET that is difficult to distinguish from noise. Hence, the precision of identifying each plastic has approximately approached 81.25% by performing the compensation based on the spectral properties even when the single substance was concealed by various elements such as the ones floating on water or while surrounded by ice.

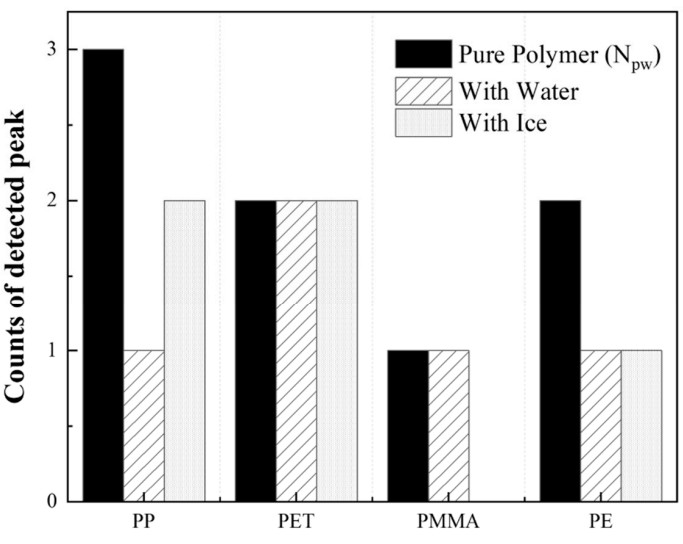

(a) Single species exposed to each environment

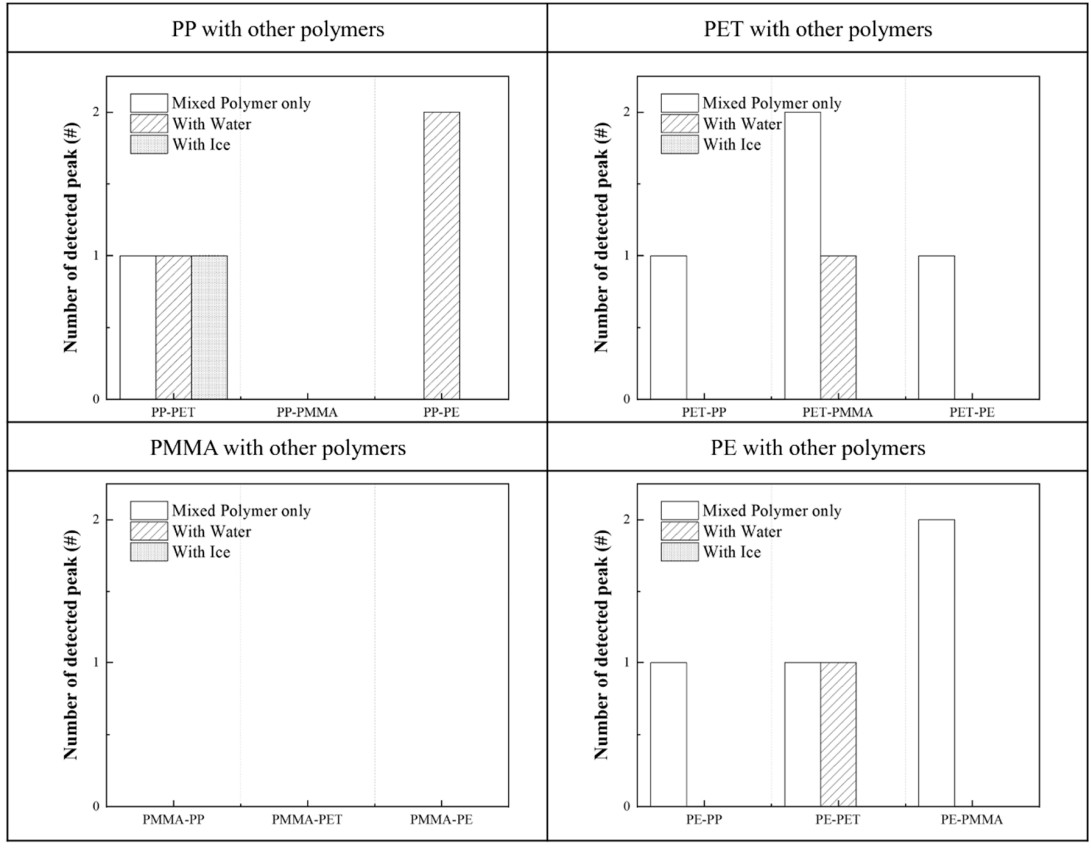

(b) Mixed species exposed to each environment

**Figure 7.** Degree of identification and classification of plastics under various conditions. (**a**) Single species and (**b**) polymer mixture.

When two different MPs are mixed under the same conditions, the identification accuracy drops sharply because the peak wavelengths are detected in similar spectral ranges. Nevertheless, other plastics except the PMMA can be detected from the given conditions, and each material is identified as shown in Figure 7b. For example, the PP and PET in the PP–PET mixture can be distinguished under all conditions but only the PP floating on the water in the PP–PE mixture is distinguished by one peak wavelength. However, any plastic mixed with the PMMA cannot be identified under all conditions. Each MP in mixtures cannot accurately be distinguished in the spectral region, 800 ∼ 1000 nm, so one needs to improve the identifying process with other techniques.

## 4. Conclusions

This research is focused on the systematic detection based on the spectroscopic properties of each plastic, PP, PET, PMMA, and PE, in the spectra range 800~1000 nm. The analysis of plastics floating on water or contained in ice can be carried out with a resolution of the spectrometer of up to 3 nm. The coding breaks down the prominent peak wavelengths in a repetitive signal gained directly from the surroundings into several Gaussian functions and isolates the unique ones of the materials by adjusting them to the initial data. This indirectly confirms the presence of the polymers distributed in the water or ice and provides additional fragmentary information about the material. However, it presents the event of a spectrum and makes the analysis difficult because the spectral signals of the materials can be mitigated and made blunt under various environmental factors. For these reasons, some materials, especially PMMA, are still poorly judged in the identification process shown in Figure 7. Nevertheless, the result shows the possibility that even very weak optical signals can be used for detecting plastics through various mathematical techniques. For instance, the spectrum of PET, 868 nm, has a reflectance that is approximately 1/8th smaller reflectance than that of the 1130 nm, whose optical characteristics are provided in another study [56]. After fitting it as the Gaussian function based on the continuum line subtraction and the LM method with the convex hull, the current procedure can characterize the weak signal as the normalized reflection. Furthermore, even if it is mixed with other polymer materials, the presence of polymers can be confirmed based on their unique properties. However, the mathematical model-based analysis can be effective in extracting the properties of the target materials from the noise-filled signals and are difficult to use in low intensity of spectra because the convex hull can be vulnerable to unexpected external noise. Recently, machine learning to solve the problem is introduced to derive results flexibly depending on situations rather than the fixed interpretations. Zhu et al. [39] extracted and identified key features of polymers with the principal component analysis (PCA) technique and support vector machine (SVM) and achieved an accuracy of 97.5% for a single polymer. Grubber et al. [57] have shown that the accuracy of detection can be increased up to 93.5% by utilizing the convolutional neutral network (CNN) technique to distinguish the size and shape of the plastic particles. As such, machine learning overcomes the limitations of the existing equipment and establishes criteria based on data of various conditions, resulting in relatively accurate results even if there is some difference. To improve the precision of detection of MPs to trace the sources and pathways of plastics, future work is intended to generate machine learning models based on the preprocessing of data to increase analysis speed and accuracy.

**Supplementary Materials:** The following are available online at https://www.mdpi.com/article/10.3390/rs13173532/s1, Algorithm S1: Convex hull, Algorithm S2: Gaussian deconvolution with Levenberg Marquardt method, Algorithm S3: Compensation for eliminating environmental effects, Algorithm S4: Polymer prediction, Figure S1: Pure polymer (PE), Figure S2: PET in water.

**Author Contributions:** Conceptualization, J.-H.K. and J.K.; methodology, J.-H.K. and J.K.; software, S.J.; validation, S.J.; formal analysis, S.J. and J.K.; investigation, S.J. and J.K.; resources, S.J. and J.K.; data curation, S.J.; writing—original draft preparation, S.J.; writing—review and editing, J.K.; visual-

ization, S.J.; supervision, J.-H.K. and J.K.; project administration, J.-H.K. and J.K.; funding acquisition, J.-H.K. and J.K. All authors have read and agreed to the published version of the manuscript.

**Funding:** This research was funded by the National Research Foundation of Korea, grant numbers NRF- No. 2020R1I1A1A01052946 and NRF-2019H1D8A2107264.

**Institutional Review Board Statement:** Not applicable.

**Informed Consent Statement:** Not applicable.

**Data Availability Statement:** The data presented in this study are available on request from the corresponding author.

**Conflicts of Interest:** The authors declare that they have no conflict of interest.

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
