# Peer review of "Detection of Microplastics in Water and Ice"

_remotesensing, doi:10.3390/rs13173532_

Round 1
Reviewer 1 Report
Microplastics in aquatic environments have been a huge environmental problem, with serious risks to water ecosystem and even human health. Using near-infrared spectra to identify microplatics in aquatic environments has large potential to determine the risk and take appropriate measures to reduce the risk. Due to the noise and disturbance by the surrounding reflected spectra, identifying microplastics has been a big challenge. In this research, the authors applied convex hull, Gaussian deconvolution and curve fitting methods to amplify and normalize reflectance. This research provides some new findings, but some minor changes are stilled.
In this study, four plastic polymers, PP, PET, PMMA and PP were analysed. More details about the rationale for choosing such four plastics are needed. Another big concern is “When two different MPs are mixed under the same conditions, the accuracy of identification is sharply reduced because the peak wavelengths are detected in a similar spectral range. ” (L362-363). I would say this is a major drawback of this method. To apply the research results more widely, authors need to propose some approaches to improve it, in a section limitation and future research.
Some minor changes are as follows:
Fig. 2 delete “OriginPro 2020 (Academic)” as this is the method/software, it is unnecessary
L229, change ; to :
L233, change ‘drag and drop’ to ‘dragged and dropped’
L266, change ‘are’ to ‘were’
Author Response
Microplastics in aquatic environments have been a huge environmental problem, with serious risks to water ecosystem and even human health. Using near-infrared spectra to identify microplatics in aquatic environments has large potential to determine the risk and take appropriate measures to reduce the risk. Due to the noise and disturbance by the surrounding reflected spectra, identifying microplastics has been a big challenge. In this research, the authors applied convex hull, Gaussian deconvolution and curve fitting methods to amplify and normalize reflectance. This research provides some new findings, but some minor changes are stilled.
In this study, four plastic polymers, PP, PET, PMMA and PP were analysed. More details about the rationale for choosing such four plastics are needed.
- Answer : The authors appreciate the reviewer for an important comment. The detailed information was included in section 1 (L41~48).
“Also, the plastics such as PE and PP mostly ….”
Another big concern is “When two different MPs are mixed under the same conditions, the accuracy of identification is sharply reduced because the peak wavelengths are detected in a similar spectral range. ” (L362-363). I would say this is a major drawback of this method. To apply the research results more widely, authors need to propose some approaches to improve it, in a section limitation and future research.
- Answer : As the comments of the review, the authors propose an approach to improve the method in future research in section 4 (L349-359).
“Recently, machine learning to solve the problem is ….”
Some minor changes are as follows:
Fig. 2 delete “OriginPro 2020 (Academic)” as this is the method/software, it is unnecessary
L229, change ; to :
L233, change ‘drag and drop’ to ‘dragged and dropped’
L266, change ‘are’ to ‘were’
- Answer : The revision was prepared by the reviewer’s comments.

Reviewer 2 Report
- In the introduction, why the near-infrared region was used to identify plastics can be written in more detail. This description didn't see the advantages of near-infrared spectroscopy.
- In this study, only the reflection spectra of various polymers are used for analysis. What is the purpose of comparing the transmission, reflection and absorption spectra of various polymers?
- The definition size of microplastics in the introduction is 100 nm-5 mm. It is recommended to select samples with smaller size for spectral determination, so as to meet the microplastics mentioned in the title
-
Misspelled line 64, H_ 2O to H2O
-
From line 99 to line 104, the figure was inconsistent with the text. It is said that the polymer floating on the water surface in figure (b) is used as a reference, but figure (b) shows the polymer exposed to ice, so how should the specific reference spectrum be defined?
-
There were two figures 1 in the full text, resulting in the wrong order of subsequent figures. Please correct the picture number.
-
In line 339-340, fig(a )indicated the peak exposed to ice, but showed the characteristic peak of water. Please adjust the order of the figure.
-
Figure 8 only theoretically distinguished the characteristic absorption peaks that can be measured by different plastic mixtures, but there is no actual measurement. It is more convincing to supplement the experiment.
Author Response
- In the introduction, why the near-infrared region was used to identify plastics can be written in more detail. This description didn't see the advantages of near-infrared spectroscopy.
- Answer : The authors appreciate the reviewer for an important comment. The description about advantages of NIR spectroscopic region was included in section 1 (L49~56).
- In this study, only the reflection spectra of various polymers are used for analysis. What is the purpose of comparing the transmission, reflection and absorption spectra of various polymers?
- Answer : As the comment of the reviewer, this study focused on the reflection spectra of various polymers for analysis. When measuring with a UV/Vis/NIR spectrometer to obtain the exact peak wavelength of the plastic used, the measurement results obtained according to the opacity of the sample are shown in the figure. Since the purpose is to confirm the location of the wavelength, the authors has corrected the content and reflected in the manuscript (L175~177).
- The definition size of microplastics in the introduction is 100 nm-5 mm. It is recommended to select samples with smaller size for spectral determination, so as to meet the microplastics mentioned in the title
- Answer : As the comment of the reviewer, the authors selected samples that are very close to 5mm, the largest limit for the size of plate-shaped plastics because it was possible to control the experimental conditions such as being exposed to and covered with the ice. The goal of this research was to identify the type of microplastics by the made coding suing the spectrum measured in the part of the NIR region under these conditions. Currently, the authors are modifying the coding to improve the precision and will provide the results of experiments by plastic pellets of less than 5 mm or pollution gases with improved coding for future work.
- Misspelled line 64, H_ 2O to H2O
- Answer : The error seems to be by MS-office. The word is corrected.
- From line 99 to line 104, the figure was inconsistent with the text. It is said that the polymer floating on the water surface in figure (b) is used as a reference, but figure (b) shows the polymer exposed to ice, so how should the specific reference spectrum be defined?
- Answer :As the reviewer recommended, the sentence and the Figure 1 are corrected.
- There were two figures 1 in the full text, resulting in the wrong order of subsequent figures. Please correct the picture number.
- Answer :The error seems to be due to auto-numbering of MS-office. The figure numbers are modified as recommended.
- In line 339-340, fig(a)indicated the peak exposed to ice, but showed the characteristic peak of water. Please adjust the order of the figure.
- Answer :The Figure (c) shows the plastics floated on water and Figure (a) and (b) are related to the ice.
- Figure 8 only theoretically distinguished the characteristic absorption peaks that can be measured by different plastic mixtures, but there is no actual measurement. It is more convincing to supplement the experiment.
- Answer : The authors appreciate the reviewer for the valuable comments. Figure 8 (newly Figure 7) shows the results obtained in the sequence described in section 3.2. After obtaining the optical spectra of the plastic mixtures contained in water and ice with an ASD FieldSpec 3 spectrometer, each substance was respectively identified using the method provided.

Reviewer 3 Report
Dear authors,
I regret that at this point I can not recommend this manuscript for publication. First of all, it needs heavy language editing, in this form it is almost unreadable, it is seriously distracting.
I find the topic interesting, but it seems to me that many aspects are only superficially handled:
- in the introduction the absorption features of water the way you presented it, shows me that you are not handling very well the matter that you are examining. Please dig more into the molecular structure of water, water phases - vapour, liquid, gel, ice and the corresponding spectral features - fundamental bands, overtones, combination bands, and very specific absorbance bands corresponding to particular water species (for example in your Introduction 940 nm is very well defined absorbance band of free water molecules), 1450 nm is a center of a very broad 1st overtone of stretching vibrations but within this band, are at around 12 to 16 water absorbance bands for example 1410 - free water, 1398 nm is water confined in the local field of ions, 1520 nm is strongly bound water, there are others for example protonated clusters, hydrogen bonded with 1, 2, 3 or 4 HB etc...
- The Material and method section contains lot of text that belong either in the introduction or in discussion section
- I do not understand why would you put reflectance, transmittance and absorbance spectra on the same graph? for me it is also strange, because, if it is remote sensing, are we not going to rely on reflectance??
- Why does it matter to distinguish which polymer it is? It is pollution, and I think the most likely scenario in practice would be a mix of all of them
- I think the true problem here is not recognition of the polymer, but detection of any of the pollutants even in minute concentrations
- I am insisting on the spectral features of water (in the text above) because you worked with pure water and ice. However, the spectra of water contains absorbance bands of all phases of water vapor, liquid, ice, and is not only influenced by temperature (heavily influenced), but also by other factors - pressure, environmental humidity, other substances..for example salt in some concentration can really produce spectral features as if it water is in ice-like state. So, I think it is important first to understand better what you try to model, and redefine what is the objective of your work.
- With the current experimental setup and current approach, I do not think the results and the solution you propose for application will be of use
Regrettably, based on the previous explanation, I can not at this point recommend the article for publication.

Author Response
The authors appreciate the reviewer for the valuable comments. Based on the comments, the revision is prepared and includes more information as well as detailed explanations and reviewed the manuscript with native English-speaking colleagues.
Dear authors,
I regret that at this point I can not recommend this manuscript for publication. First of all, it needs heavy language editing, in this form it is almost unreadable, it is seriously distracting.
I find the topic interesting, but it seems to me that many aspects are only superficially handled:
- in the introduction the absorption features of water the way you presented it, shows me that you are not handling very well the matter that you are examining. Please dig more into the molecular structure of water, water phases - vapour, liquid, gel, ice and the corresponding spectral features - fundamental bands, overtones, combination bands, and very specific absorbance bands corresponding to particular water species (for example in your Introduction 940 nm is very well defined absorbance band of free water molecules), 1450 nm is a center of a very broad 1st overtone of stretching vibrations but within this band, are at around 12 to 16 water absorbance bands for example 1410 - free water, 1398 nm is water confined in the local field of ions, 1520 nm is strongly bound water, there are others for example protonated clusters, hydrogen bonded with 1, 2, 3 or 4 HB etc...
- Answer : The authors appreciate the reviewer for an important comment. The detailed information was included in section 1 (L49~67). The research focused on the wavelength between 750 and 1000nm because of the relatively small effect of the temperature.
- The Material and method section contains lot of text that belong either in the introduction or in discussion section
- Answer : As indicated by the reviewer, the part was a move to the discussion section. The revision was prepared by the reviewer’s comments.
- I do not understand why would you put reflectance, transmittance and absorbance spectra on the same graph? for me it is also strange, because, if it is remote sensing, are we not going to rely on reflectance??
- Answer : As the comment of the reviewer, this study focused on the reflection spectra of various polymers for analysis. When measuring with a UV/Vis/NIR spectrometer to obtain the exact peak wavelength of the plastic used, the measurement results obtained according to the opacity of the sample are shown in the figure. Since the purpose is to confirm the location of the wavelength, the authors has corrected the content and reflected in the manuscript (L175~177).
- Why does it matter to distinguish which polymer it is? It is pollution, and I think the most likely scenario in practice would be a mix of all of them
- Answer : The authors appreciate the reviewer for an important comment. The detailed information was included in section 1 (L41~48).
“Also, the plastics such as PE and PP mostly ….”
- I think the true problem here is not recognition of the polymer, but detection of any of the pollutants even in minute concentrations
- Answer : The authors first considered tracing the sources and pathways of plastics in water and ice in urban and rural areas in South Korea. After the authors improve the precision of the analysis method, the study will be extended to detect microplastics in minute concentrations as the reviewer’s comment.
- I am insisting on the spectral features of water (in the text above) because you worked with pure water and ice. However, the spectra of water contains absorbance bands of all phases of water vapor, liquid, ice, and is not only influenced by temperature (heavily influenced), but also by other factors - pressure, environmental humidity, other substances..for example salt in some concentration can really produce spectral features as if it water is in ice-like state. So, I think it is important first to understand better what you try to model, and redefine what is the objective of your work.
- Answer : As the comment of the reviewer, the detailed information was included in the content (L61-63, L86-88). The authors did not explain the detailed experimental conditions, but the absorption peaks of water and ice were continuously monitored with the temperature change. The temperatures of water and ice were uniformly maintained to neglect the effect of temperature during the experiment. Based on this experiment, more plastics and other environmental conditions will be tested to improve to detection ranges as well as classification.
- With the current experimental setup and current approach, I do not think the results and the solution you propose for application will be of use
Regrettably, based on the previous explanation, I cannot at this point recommend the article for publication.
- Answer : The authors appreciate the reviewer’s critical comment. The authors also knew the limited scoped of the experiment, however, the primary goal of the research is the feasibility study to detect and classify the microplastics debris on earth, especially on water near rivers and oceans. In this study, the authors tested the basic classification of microplastic with a simple experimental setup, resulting in 81% classification accuracy with four different polymers. Based on this experiment, more plastics and other environmental conditions will be tested to improve the detection ranges as well as classification. Also, other setups with a collimated system will assist to improve the detection.

Round 2
Reviewer 2 Report
1. L21-L22, the peak description of PET and PMMA is too vague and easy to be confused.
2. L79- The use of 50 ×50 mm2 polymer is not clearly explained in the manuscript, and the measurement of this size of polymers is not necessary.
3.In this manuscript, the experiment only distinguishes between two polymers after mixing. Please analyze the situation when three or four polymers are mixed.
4. PMMA polymers cannot be identified with ice, so some solutions or future prospects need to be put forward.
5. Please carefully check the spelling mistakes in the manuscript to improve the language readability.
Author Response
The authors again appreciate the reviewer for the valuable comments. Based on the comments, the revision is prepared and includes more information as well as detailed explanations.
- L21-L22, the peak description of PET and PMMA is too vague and easy to be confused.
- Answer : As the comment of the reviewer, the sentence is changed.
“And unique peak wavelengths with the meaningful measure at 868 and 907 nm for the PET and 887 nm for the PMMA.”
- L79- The use of 50 ×50 mm2 polymer is not clearly explained in the manuscript, and the measurement of this size of polymers is not necessary.
- Answer : The authors appreciate the reviewer for the kind comments. As mentioned Figure 3 and 3.1 Peak wavelengths of plastic polymers, 50×50 mm2polymers were used to measure the peak wavelengths of polymers with the Lambda 750 UV/Vis/NIR spectrometer.
“The peak wavelengths of the PP, PET, PMMA, and PE, in a rectangular cross-section with 5 mm in thickness, were measured using the Lambda 750 UV/Vis/NIR spectrometer (PerkinElmer, USA) from 800 to 1000 nm as followed”
In this manuscript, the experiment only distinguishes between two polymers after mixing. Please analyze the situation when three or four polymers are mixed.
- Answer : The authors appreciate the reviewer for the valuable comments. The primary goal of the research is the feasibility study to detect and classify the microplastics debris with the in-house program. In this study, the authors tested the basic classification of microplastic with a simple experimental setup. Based on this experiment and with improving the in-house program, the study will be conducted on the identification of each substance in the various plastic mixtures.
- PMMA polymers cannot be identified with ice, so some solutions or future prospects need to be put forward.
- Answer : As the comment of the reviewer, the authors are mentioned the future work on L349~359 for improving the precision of identification and classification of plastics including PMMA polymer.
- Please carefully check the spelling mistakes in the manuscript to improve the language readability.
- Answer : Based on the comments, the revision is prepared and reviewed the manuscript with English editing.

Reviewer 3 Report
Dear authors,
I am very satisfied with the improvements you performed, it is clear that you acted upon all the requests by the reviewers and greatly improved the manuscript. The points that still need improvements are related to the English language and text formatting, and the quality of images. Language is really, still very poor and requires help, the newly introduced paragraphs have many occurrences of the words joined together. And images are very blurry. Please improve this.
Best regards
Author Response
Dear authors,
I am very satisfied with the improvements you performed, it is clear that you acted upon all the requests by the reviewers and greatly improved the manuscript. The points that still need improvements are related to the English language and text formatting, and the quality of images. Language is really, still very poor and requires help, the newly introduced paragraphs have many occurrences of the words joined together. And images are very blurry. Please improve this.
Best regards
- Answer : The authors really appreciate the reviewer for the kind and valuable comments again. Based on the comments, the revision is prepared and reviewed the manuscript with English editing.
